# Residential Radon Exposure and Cigarette Smoking in Association with Lung Cancer: A Matched Case-Control Study in Korea

**DOI:** 10.3390/ijerph17082946

**Published:** 2020-04-24

**Authors:** Eung Joo Park, Hokyou Lee, Hyeon Chang Kim, Seung Soo Sheen, Sang Baek Koh, Ki Soo Park, Nam Han Cho, Cheol-Min Lee, Dae Ryong Kang

**Affiliations:** 1Department of Precision Medicine & Biostatistics, Yonsei University Wonju College of Medicine, Wonju 26426, Korea; puck052@naver.com; 2Center of Biomedical Data Science, Yonsei University Wonju College of Medicine, Wonju 26426, Korea; 3Department of Preventive Medicine, Yonsei University College of Medicine, Seoul 03722, Korea; HOKYOU.LEE@yuhs.ac (H.L.); HCKIM@yuhs.ac (H.C.K.); 4Department of Internal Medicine, Yonsei University College of Medicine, Seoul 03722, Korea; 5Cardiovascular and Metabolic Disease Etiology Research Center, Yonsei University College of Medicine, Seoul 03722, Korea; 6Department of Pulmonary and Critical Care Medicine, Ajou University School of Medicine, Suwon 16499, Korea; sssheen@ajou.ac.kr; 7Department of Preventive Medicine, Yonsei University Wonju College of Medicine, Wonju 26426, Korea; kohhj@yonsei.ac.kr; 8Department of Preventive Medicine, School of Medicine, Gyeongsang National University, Jinju 52828, Korea; parkks@gnu.ac.kr; 9Department of Preventive Medicine, Ajou University School of Medicine, Suwon 16499, Korea; chnaha@ajou.ac.kr; 10Department of Chemical and Biological Engineering, Seokyeong University, Seoul 02713, Korea; cheolmin@skuniv.ac.kr

**Keywords:** radon, cigarette smoking, lung cancer

## Abstract

Residential radon exposure and cigarette smoking are the two most important risk factors for lung cancer. The combined effects thereof were evaluated in a multi-center matched case-control study in South Korea. A total of 1038 participants were included, comprising 519 non-small cell lung cancer cases and 519 age- and sex- matched community-based controls. Residential radon levels were measured for all participants. Multivariate logistic regression was used to calculate odds ratios (OR) for lung cancer according to radon exposure (high ≥ 100 Bq/m^3^ vs. low < 100 Bq/m^3^), smoking status, and combinations of the two after adjusting for age, sex, indoor hours, and other housing information. The median age of the participants was 64 years, and 51.3% were women. The adjusted ORs (95% confidence intervals [CIs]) for high radon and cigarette smoking were 1.56 (1.03–2.37) and 2.53 (1.60–3.99), respectively. When stratified according to combinations of radon exposure and smoking status, the adjusted ORs (95% CIs) for lung cancer in high-radon non-smokers, low-radon smokers, and high-radon smokers were 1.40 (0.81–2.43), 2.42 (1.49–3.92), and 4.27 (2.14–8.52), respectively, with reference to low-radon non-smokers. Both residential radon and cigarette smoking were associated with increased odds for lung cancer, and the difference in ORs according to radon exposure was much greater in smokers than in non-smokers.

## 1. Introduction

An aggressive cancer, lung cancer is the most common cause of cancer death worldwide [1]. While prognoses of lung cancer at advanced stages remain disappointing [2], survival rates for early localized disease are often promising [3], although early detection of lung cancer is uncommon. Therefore, current preventive strategies focus on controlling environmental hazards or routine radiologic screening of individuals at high risk for lung cancer [4].

The two most important environmental contributors to lung cancer development are cigarette smoking and radon exposure. Of these, exposure to radon indoors has garnered greater interest as a risk factor for lung cancer, as radon is a colorless and odorless gas that is ubiquitous in rocks and soils and, thus, can accumulate in buildings [5]. The association between radon exposure and lung cancer has been widely reported [6,7,8] and the combined effect of radon and tobacco smoke is thought to be higher than additive [9,10]. However, studies on the combined effect of radon and smoking on lung cancer have primarily focused on exposure to radon at high concentration sites, such as uranium mines [11,12,13]. Accordingly, we aimed to assess the interaction between residential radon exposure and cigarette smoking in association with lung cancer in a matched case-control study in Korea.

## 2. Methods

### 2.1. Study Design and Participants

The Korea-Integrated Radon Exposure Epidemiology Statistics (K-iREES) study enrolled a total of 6582 individuals between October 2015 and March 2018 from seven tertiary hospitals and four community-based cohorts (Figure 1). The hospital-based participants were recruited from Severance Hospital, Seoul; Asan Hospital, Seoul; St. Mary’s Hospital, Seoul; Ajou University Hospital, Suwon; Wonju Severance Hospital, Wonju; Gyeongsang University Hospital, Changwon; and Pusan University Hospital, Yangsan. The community-based participants were recruited from the Cardiovascular and Metabolic Diseases Etiology Research Center (CMERC), Seoul [14]; the Ansung-Ansan Korean Genome and Epidemiology Study (KoGES) [15]; the Namgaram cohort [16], Gyeongnam; and the Wonju-Pyeongchang KoGES [15]. The study regions were selected to include various regional radon exposure levels according to data obtained from the National Institute of Environmental Research (2011–2016). The selected study regions and their corresponding exposure levels are depicted in Figure 2, with correction for seasonal variations.

From the study hospitals, patients aged 19 to 80 years who had been diagnosed with non-small cell lung cancer (NSCLC) stage I to IIIa were included. The controls were selected from community-based cohort participants aged 19 to 80 years who had no known diagnosis of lung cancer. All participants had lived in their homes for 2 years or longer. A total of 1343 individuals, including 526 hospital-based and 817 community-based participants, had radon measurements taken in their homes. For each hospital-based lung cancer patient, a community-based control was matched for sex and age (<65 or ≥65 years), and 1:1 sampled using SAS *proc surveyselect*. Finally, 519 cases and 519 matched controls were analyzed. The study protocol was approved by the Institutional Review Board of Yonsei University College of Medicine (CR315030).

### 2.2. Measurement of Residential Radon Levels

Residential radon levels were measured at two locations in each study home where individuals tend to spend most of their time: the living room and the bedroom. Alpha-track detectors (Raduet Model RSV-8; Radosys Ltd., Budapest, Hungary) were used as a passive radon measuring device. The measuring devices were positioned away from household electrical appliances, windows, or sealed drawers. The measurements were made over 3 months, and the average of measurements at both locations in the house was taken. Given that indoor radon levels are highest in the winter and lowest in the summer, seasonal corrections were made with average temperature, wind speed, and other factors taken into consideration [17]. The residential radon levels were dichotomized into high (≥100 Bq/m^3^) or low (<100 Bq/m^3^) according to World Health Organization reference data [18].

### 2.3. Smoking History and Covariables

The K-iREES study was designed to investigate factors associated with radon exposures and related health problems. Questionnaires were used to identify demographics, health-related behaviors, such as cigarette smoking, and the characteristics of individual homes, including indoor cracks, ventilation, housing types, construction year, etc. Sleeping hours was also considered, with 70 percent of the time spent breathing through the nose during sleep or rest [19]. Cigarette smoking was defined as having smoked five or more packs in a lifetime. Second-hand smoking was defined as living together with or working in proximity to a current smoker [20,21]. Green area corresponds to forest and grassland area; agricultural space, such as rice fields, is not included in green area [22].

### 2.4. Statistical Analysis

Participant characteristics are reported as a mean ± standard deviation, median [interquartile range], or frequency (percent). Intergroup comparisons were conducted using *t*-tests for continuous variables and χ2-tests for categorical variables. We used multivariate conditional logistic regression to calculate odds ratios (OR) and 95% confidence intervals (CI) for lung cancer according to residential radon exposure (high vs. low), smoking status, and combinations of the two (low-radon dwelling non-smokers [reference], high-radon dwelling smokers, low-radon dwelling smokers, and high-radon dwelling smokers), after adjusting for second-hand smoking, sleeping hours, indoor hours, housing type, floor, presence of cracks, and green ratio [22]. All analyses were performed using SAS version 9.4 (SAS Institute Inc., Cary, NC, USA). Map-visualization of radon levels was computed using R version 3.4.3 (R Foundation for Statistical Computing, Vienna, Austria).

## 3. Results

### 3.1. Participant Characteristics

Descriptive statistics of the 519 hospital-based lung cancer cases and the 519 age- and sex-matched community-based controls are reported in Table 1. In both the case and control groups, the median age was 64 years, and 51.3% were women. Mean residential radon levels were 65.46 Bq/m^3^ and 73.75 Bq/m^3^ (*p* = 0.013) in the case and control groups, respectively. Among the cases and controls, the proportions of individuals exposed to high levels of residential radon (≥100 Bq/m^3^) were 13.7% and 17.7% (*p* = 0.007); smokers comprised 42.8% and 34.9% (*p* = 0.009); and second-hand smokers accounted for 46.1% and 21.2% (*p* < 0.001), respectively. Participants in the case group reported longer sleeping and indoor hours and were more likely to live in apartments or other multi-family houses, with a lower green ratio (all *p* < 0.001), although with similar building ages, than participants in the control group.

### 3.2. Residential Radon and Cigarette Smoking on Lung Cancer

In conditional logistic regression adjusted for second-hand smoking, sleeping and indoor hours, housing type and floor, house cracks, and green ratio, the ORs (95% CIs) for high radon, cigarette smoking and heavy-smoker were 1.56 (1.03–2.37), 2.53 (1.60–3.99), 5.56(3.31–9.35) respectively (Table 2). When stratified by combinations of radon exposure and smoking status (low-radon non-smokers [reference], high-radon smokers, low-radon smokers, and high-radon smokers), the difference in ORs for lung cancer by radon exposure was much greater in smokers than in non-smokers. That is, with low-radon non-smokers as the reference group, the adjusted ORs (95% CIs) for lung cancer were 1.40 (0.81–2.43), 2.42 (1.49–3.92), and 4.27 (2.14–8.52) in high-radon non-smokers, low-radon smokers, and high-radon smokers, respectively. Similar findings were observed when we used conventional, instead of conditional, logistic regression (Table 2).

Furthermore, we repeated the analysis with tobacco smoke-exposure reclassified into smoke-free group (neither smoking nor being exposed to second-hand smoke) and smoke-exposed group (active smoking and/or being exposed to second-hand smoke). Compared with the low-radon smoke-free group, the adjusted ORs for lung cancer in high-radon smoke-free, low-radon smoke-exposed, high-radon smoke-exposed groups were 1.01 (0.49–2.07), 2.39 (1.48–3.87), and 4.93 (2.57–9.45) from a conditional logistic model and 1.04 (0.51–2.13), 2.41 (1.49–3.89), and 4.65 (2.44–8.88) from a conventional logistic model, respectively (Table 3). 

Finally, we checked the robustness of our data using a lower radon cut-off value of 74 Bq/m^3^ [23]. The adjusted ORs were 1.55 (1.02–2.34), 2.39 (1.45–3.95), and 4.16 (2.29–7.57) in high-radon non-smokers, low-radon smokers, and high-radon smokers, respectively (Appendix A), and were comparable with ORs from the main analyses.

## 4. Discussion

In this matched case-control study, we discovered significant associations for lung cancer with residential radon exposure, with cigarette smoking, and with combinations of the two. Residential radon exposure and cigarette smoking were synergistically associated with a greater odds for lung cancer. Although addictive interaction (*p* = 0.344) and multiplicative interaction (*p* = 0.367) did not reach statistical significance, the difference in ORs for lung cancer according to radon exposure was much greater in current smokers than in non-smokers. Such trend was more pronounced when environmental smoking was taken into account. In this regard, for both smoking- and radon-related lung cancer risk, the most important risk reduction strategy would be smoking cessation and avoidance of environmental tobacco smoke. Conversely, among active or secondhand smokers, residential radon assessment and control should constitute a significant portion of lung cancer preventive measures, in addition to efforts supporting cessation and avoidance.

Even at concentrations far below the official guidance level, radon can lead to a 2.5 times increase in lung cancer risk. Furthermore, synergies found between smoking and radon can be useful in writing public health recommendations [24]. The analysis was conducted based on 1pCi/litter. We conducted the analysis based on WHO recommendation standard of 100Bq/m^3^ and 74Bq/m^3^.

Compared to the subjects studied in Spain, there are more packs of cigarettes consumed per year in Korea. Most of the subjects smoked a pack of cigarettes a day. The survey did not reveal the exact amount of cigarette consumption [25]. In the case of Heavy Smoker, the OR value was 3.38 (1.35–8.47) (*p* = 0.009) based on Radon 100Bq/m^3^. In the case of smoking, additional research is need on Never Smoker because it affects lung cancer more than Radon. Odds ratio were lower due to the higher proportion of non-smoking subjects and women than previous studies.

The carcinogenicity of radon and cigarette smoke may involve various mechanisms, including generation of DNA-reactive products, chromosomal instability and aberrations, and mutations of tumor-suppressor genes [26]. However, the current literature is, as of yet, inconsistent on mutation “hotspots” or unique cytogenetic markers associated with radon-related carcinogenicity or its interactions with tobacco smoke [27]. Some in vitro studies have suggested a synergistic increase of chromosomal aberrations and possibly a higher susceptibility to radon exposure in lymphocytes of smokers [28,29]. It has also been proposed that radon progeny may attach to tobacco smoke aerosols and increase potential doses to target organs [30,31]. Further molecular and cytogenetic studies are needed to elucidate the mechanism underlying the observed synergism between low-dose radon and smoking in association with lung cancer.

Epidemiologic evidence of interactions between radon exposure and cigarette smoking and their effects on lung cancer has been described in a number of studies [10,32,33]. However, many of these studies included persons exposed to a high doses of radon, such as those face by uranium miners [34,35,36] Considering the non-linear dose-response relationship between radon and lung cancer, the modifying effect of low-dose radon on the smoking-lung cancer relationship may not be extrapolated from uranium miner results. In this study, we evaluated the interaction between residential radon and cigarette smoking, and our findings hold notable implications in lung cancer risk assessment and preventive measures. Furthermore, this is the first study in Korea to describe interactions between residential radon and cigarette smoking in association with lung cancer.

Our study has several limitations. First, the case-control design precludes causal inference between the exposure variables and lung cancer. Second, although we incorporated a matched case-control design and further adjustments for other imbalances, residual and unmeasured confounding may exist. Third, the number of female smokers in our study was too small for sex-specific analyses to be possible. Fourth, recall bias in smoking history is also possible. Finally, histopathologic subtypes of NSCLC were not differentiated in our study. Notwithstanding, this study also has some notable strengths. Foremost, we used individual-level residential radon measurements rather than ecologic data. Moreover, the cases and controls were gathered from multiple centers and cohorts of different geographic locations with varying regional radon levels. Therefore, our findings may provide some generalizability on radon exposure patterns and their associations with lung cancer in Korea.

In conclusion, we found both residential radon and cigarette smoking to be associated with increased odds for lung cancer, and the difference in ORs according to radon exposure was much greater in smokers than in non-smokers. Therefore, preventive strategies targeting radon-related lung cancer should emphasize, in addition to radon-reducing repairs and ventilation, both smoking cessation and withdrawing from second-hand smoking.

## Figures and Tables

**Figure 1 ijerph-17-02946-f001:**
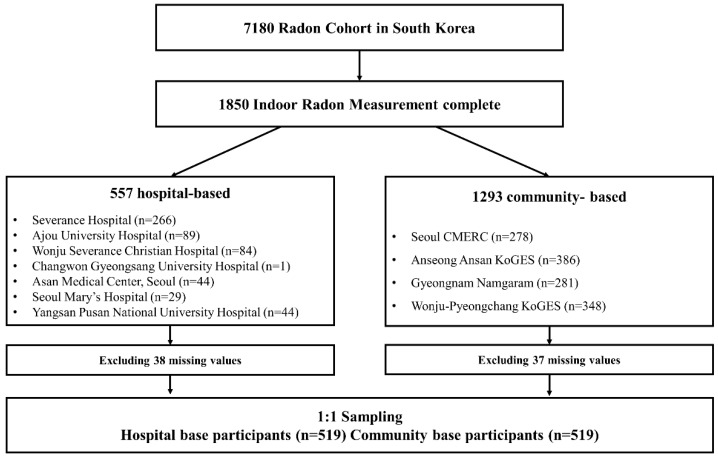
Flowchart of the study participants.

**Figure 2 ijerph-17-02946-f002:**
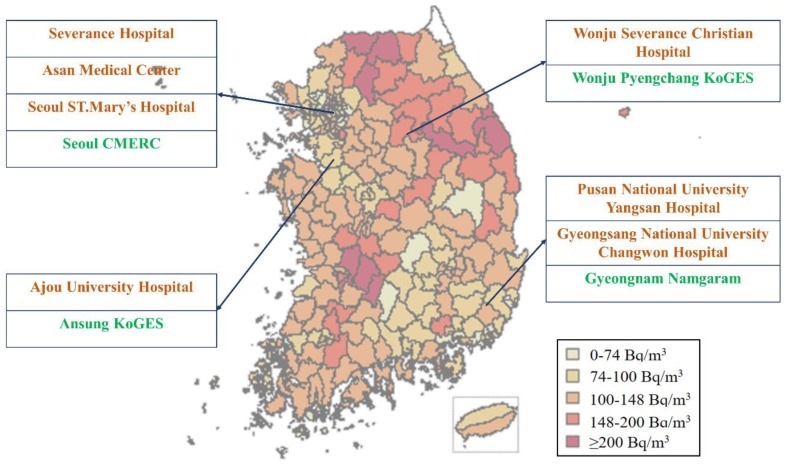
Map of the study area and research sites. Regional indoor radon levels were obtained from the National Institute of Environmental Research (2011–2016). Fill colors correspond to radon levels in five categories. Stars designate the locations of study sites.

**Table 1 ijerph-17-02946-t001:** Characteristics of the study participants.

Variables	Case (N = 519)	Control (N = 519)	*p* Value
Age, years	64 [57–72]	64 [59–72]	0.116
Sex, n (%)			N/A
Male	253 (48.75)	253 (48.75)	
Female	266 (51.25)	266 (51.25)	
Residential radon *, Bq/m^3^	65.46 ± 46.71	73.75 ± 60.21	0.013
	48.32 [34.43–73.61]	55.06 [37.71–82.78]	<0.001
High-radon dwelling ^†^, n (%)	71 (13.68)	92 (17.73)	0.007
Cigarette smoking, n (%)	222 (42.77)	181 (34.87)	0.009
Tobacco consumption, n (%)			<0.001
Never-smokers	297 (57.23)	338 (65.13)	
Light smokers (1–100 pack-years)	9 (1.73)	7 (1.35)	
Moderate smokers (100–365 pack-years)	32 (6.17)	34 (6.55)	
Heavy smokers (over 365 pack-years)	178 (34.30)	85 (16.38)	
Non-response	3(0.58)	55(10.60)	
Second-hand smoking, n (%)	239 (46.05)	110 (21.19)	<0.001
Sleeping hours	7.20 ± 1.83	6.76 ± 1.44	<0.001
Indoor hours	15.88 ± 4.39	14.17 ± 3.69	<0.001
Housing type, n (%)			<0.001
Single-family house	178 (34.30)	373 (71.87)	
Apartment	180 (34.68)	68 (13.10)	
Other multi-family dwelling	161 (31.02)	78 (15.03)	
Floor of residence	4.76 ± 5.09	2.63 ± 3.69	<0.001
Presence of house crack, n (%)	120 (23.12)	145 (27.94)	0.075
Construction year	1996 [1990–2003]	1997 [1987–2005]	0.638
Green ratio	48.09 ± 21.09	56.89 ± 18.54	<0.001

* Corrected for seasonal variations. ^†^Residential radon ≥ 100 Bq/m^3^.

**Table 2 ijerph-17-02946-t002:** Associations of residential radon exposure and cigarette smoking with lung cancer.

Variables	Case, n	Control, n	Conditional Logistic Regression		Conventional Logistic Regression	
			OR (95% CI) *	*p*-Value	OR (95% CI) ^†^	*p*-Value
Residential radon						
Low (< 100 Bq/m^3^)	448	427	1.00 (reference)		1.00 (reference)	
High (≥ 100 Bq/m^3^)	71	92	1.56 (1.03–2.37)	0.037	1.52 (1.00–2.31)	0.048
Cigarette smoking						
Non-smokers	297	338	1.00 (reference)		1.00 (reference)	
Smokers	222	181	2.53 (1.60–3.99)	<0.001	2.50 (1.59–3.94)	<0.001
Tobacco consumption						
Never-smoker	297	338	1.00 (reference)		1.00 (reference)	
Light smokers	9	7	3.05 (0.81–11.43)	0.739	2.47 (0.68–8.56)	0.797
Moderate smokers	32	34	2.65 (1.32–5.30)	0.934	2.03 (1.11–3.71)	0.847
Heavy smokers	178	85	5.56 (3.31–9.35)	<0.001	4.24 (2.92–6.15)	<0.001
Radon and smoking						
Low-radon non-smokers	262	282	1.00 (reference)		1.00 (reference)	
High-radon non-smokers	35	56	1.40 (0.81–2.43)	0.231	1.40 (0.81–2.44)	0.230
Low-radon smokers	186	145	2.42 (1.49–3.92)	<0.001	2.42 (1.50–3.91)	<0.001
High-radon smokers	36	36	4.27 (2.14–8.52)	<0.001	4.02 (2.03–7.97)	<0.001

* Conditional logistic regression was adjusted for second-hand smoking, sleeping hours, indoor hours, housing type, floor, presence of house cracks, and green ratio. CI, confidence interval; OR, odds ratio. ^†^ Conventional logistic regression was further adjusted for age and sex.

**Table 3 ijerph-17-02946-t003:** Associations of residential radon and tobacco smoke exposure with lung cancer.

Variables	Case, n	Control, n	Conditional Logistic Regression		Conventional Logistic Regression	
			OR (95% CI) *	*p*-Value	OR (95% CI) ^†^	*p*-Value
Residential radon						
Low (<100 Bq/m^3^)	448	427	1.00 (reference)		1.00 (reference)	
High (≥100 Bq/m^3^)	71	92	1.56 (1.03–2.37)	0.037	1.52 (1.00–2.31)	0.048
Smoke exposure						
Smoke-free	122	254	1.00 (reference)		1.00 (reference)	
Smoke-exposed	397	265	2.67 (1.69–4.21)	<0.001	2.64 (1.68–4.17)	<0.001
Radon and smoke exposure						
Low-radon smoke-free	109	204	1.00 (reference)		1.00 (reference)	
High-radon smoke-free	13	50	1.01 (0.49–2.07)	0.956	1.04 (0.51–2.13)	0.919
Low-radon smoke-exposed	339	223	2.39 (1.48–3.87)	<0.001	2.41 (1.49–3.89)	<0.001
High-radon smoke-exposed	58	42	4.93 (2.57–9.45)	<0.001	4.65 (2.44–8.88)	<0.001

* Adjusted for second-hand smoking, sleeping hours, indoor hours, housing type, floor, presence of house cracks, and green ratio. CI, confidence interval; OR, odds ratio. ^†^ Conventional logistic regression was further adjusted for age and sex.

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
