# Peer review of "Residential Radon Exposure and Cigarette Smoking in Association with Lung Cancer: A Matched Case-Control Study in Korea"

_ijerph, 2020, doi:10.3390/ijerph17082946_

Round 1

Reviewer 1 Report

General Comments: This manuscript is a case control study conducted in South Korea from 2015-2018. The authors are interested in combined efforts of low dose radon exposure and tobacco use/exposure and their impact on lung cancer. This manuscript is novel as it provides a well justified research design and in home radon assessment. 

Specific points to include:

Methods: Please provide a reference for information on the validity and reliability of the Raduct Model RSV-8 alpha track detector. An additional reference is also needed as to why smoking was defined as 5 or more packs in a lifetime. I am unfamiliar with this definition and, as such, including a reference or two of previous studies that have used this definition is a must. Please add a one sentence rationale for why you included sleeping hours as a covariate and include a reference. Finally, please share your definition for the covariate "green ratio" and a reference for those of your reader that may be unfamiliart with this concept.

Discussion: You may consider adding a paragraph on public awareness of radon/tobacco and their synergistic risk. If you so choose, these references may be of assistance.

Butler, K. M., Huntington-Moskos, L., Rayens, M. K., Wiggins, A. T., & Hahn, E. J. (2019). Perceived synergistic risk for lung cancer after environmental report-back study on home exposure to tobacco smoke and radon. Am J Health Promot, 33(4), 597-600. doi: 10.1177/0890117118793886

Hahn, E. J., Wiggins, A. T., Rademacher, K., Butler, K. M., Huntington-Moskos, L., & Rayens, M. K. (2019). FRESH: Long-term outcomes of a randomized trial to reduce radon and tobacco smoke in the home. . Preventing Chronic Disease: Public Health Research, Practice, and Policy (special issue, Health Care Systems, Public Health, and Communities: Population Health Improvements). 16, E127. doi: 10.5888/pcd16.180634

Huntington-Moskos, L., Rayens, M. K., Wiggins, A., & Hahn, E. J. (2016). Radon, secondhand smoke and children in the home: Creating a teachable moment for lung cancer prevention. Public Health Nursing. doi: 10.1111/phn.12283

Lantz, P. M., Mendez, D., & Philbert, M. A. (2013). Radon, smoking, and lung cancer: The need to refocus radon control policy. American Journal of Public Health, 103(3), 443-447. doi: 10.2105/

National Research Council. (1999). Health effects of exposure to radon: BEIR VI. Washington DC: The National Academies Press.

I appreciate this manuscript and the important contribution to the literature for 1) Radon, Tobacco, Synergistic Risk and 2) overall Environmental Health.

Author Response

Methods: Please provide a reference for information on the validity and reliability of the Raduct Model RSV-8 alpha track detector.
(Answer) We followed the Korean Research Institute of Standards and Science (KRISS) certified radon measurement method is used as a standard in Korea.

An additional reference is also needed as to why smoking was defined as 5 or more packs in a lifetime. I am unfamiliar with this definition and, as such, including a reference or two of previous studies that have used this definition is a must.

(Answer) Smoking status was defined using the National Health Interview Survey (NHIS). Reference has been added. The exact definitions are as follows: The standard NHIS current smoking definition (hereafter simply termed the “NHIS definition”) has comprised of two questions since 1965 (J. Madans, NCHS, personal communication, Nov. 10, 2011), with the present wording in use since 1992. The first question, asked of all respondents, is “have you smoked at least 100 cigarettes in your entire life?” Respondents answering “yes” are classified as ever smokers, and those who answer “no” are classified as never smokers and excluded from subsequent cigarette use questions. Ever smokers are then asked a second question: “do you now smoke cigarettes every day, some days or not at all?” Respondents who answer “every day” or “some days” are classified as current smokers

 Please add a one sentence rationale for why you included sleeping hours as a covariate and include a reference.
(Answer) We added the contents to the manuscript as requested. "Sleeping hours were also condoned, with 70 percent of the time spent breaking through the nose drying sleep or rest."

Finally, please share your definition for the covariate "green ratio" and a reference for those of your reader that may be unfamiliart with this concept.
(Answer) We added the contents to the manuscript as requested. " Green area corresponds to forest and grassland area; agricultural space, such as rice fields, is not included in green area."

Discussion: You may consider adding a paragraph on public awareness of radon/tobacco and their
synergistic risk. If you so choose, these references may be of assistance.
(Answer)Thank you for your opinion. However we're working on a paper in another study. The study is about public awareness of Radon and the burden of disease.

Reviewer 2 Report

A very good study and clearly written paper.

my only concerns which should be addressed in discussion or preferably with a little re analysis are;

  1. would all NSCLC cases from the communities in the study have been referred to one of the referral hospitals in the study?
  2. Although it is stated (p9) that a test for multiplicative  interaction was done and found non significant, this should be amplified and ideally a model with smoking, radon and radon X smoking presented.
  3. why choose 5 packs in a lifetime as smoker criterion and if packs and duration are known why not use a quantitative smoking variable?
  4. why was quantitative radon variable not used?

Author Response

  1. would all NSCLC cases from the communities in the study have been referred to one of the referral hospitals in the study?
    NSCLC case
    (Answer) All NSCLC cases in the communities under study were referred to one of the consigned hospitals.
  2. Although it is stated (p9) that a test for multiplicative interaction was done and found non significant, this should be amplified and ideally a model with smoking, radon and radon X smoking presented.
    (Answer) Thank you for your opinion. We checked the Multiplinary interaction but it was statistically meaningless. Logistic regression was performed by generating categorical values for Radon/Tabacco.
  3. why choose 5 packs in a lifetime as smoker criterion and if packs and duration are known why not use a quantitative smoking variable?
    (Answer)Smoking status was defined using the National Health Interview Survey (NHIS). Reference has been added. The exact definitions are as follows: The standard NHIS current smoking definition (hereafter simply termed the “NHIS definition”) has comprised of two questions since 1965 (J. Madans, NCHS, personal communication, Nov. 10, 2011), with the present wording in use since 1992. The first question, asked of all respondents, is “have you smoked at least 100 cigarettes in your entire life?” Respondents answering “yes” are classified as ever smokers, and those who answer “no” are classified as never smokers and excluded from subsequent cigarette use questions. Ever smokers are then asked a second question: “do you now smoke cigarettes every day, some days or not at all?” Respondents who answer “every day” or “some days” are classified as current smokers
  4. why was quantitative radon variable not used?
    (Answer) We focused our research on long-term residence in high radon and low radon, based on WHO radon criterion 100Bq/m3.